# Leaching Kinetics of Y and Eu from Waste Phosphors under Microwave Irradiation

**Delong Yang [1], Mingming Yu [1,2,\*], Yunqi Zhao [1], Mingyu Cheng [1] and Guangjun Mei [3,\*]**

1   School of Resources and Environmental Engineering, Jiangxi University of Science and Technology, Ganzhou 341000, China
2   Collaborative Innovation Center for Development and Utilization of Rare Metal Resources, Ganzhou 341000, China
3   School of Resources and Environmental Engineering, Wuhan University of Technology, Wuhan 430070, China
\*   Correspondence: mingmy1990@163.com (M.Y.); guangjunmei@whut.educn (G.M.)

**Abstract:** Waste fluorescent powder contains a large amount of rare earth elements, which have a high value for recovery and utilization. In order to achieve the rapid and efficient leaching of rare earth elements in these waste phosphors, microwave-assisted leaching of rare earth elements Y and Eu from the waste phosphor with hydrochloric acid was studied. The maximum leaching rates of Y (99.84%) and Eu (89.82%) were obtained at 600 W microwave power, 60 min microwave radiation time at 60 °C. The leaching kinetics showed that the microwave leaching process of Y and Eu conforms to the chemical reaction control model, and the apparent activation energy is 25.30 kJ/mol and 24.78 kJ/mol. Compared with the conventional heating method, the microwave leaching process can obviously reduce the reaction activation energy, shorten the reaction time, and achieve the rapid and efficient leaching of rare earth elements in the waste phosphors.

**Keywords:** waste phosphors; microwave irradiation; leaching kinetics; rare earth





## 1. Introduction

As an important strategic resource, the application of rare earths is becoming more and more extensive, which leads to the increasing demand of rare earth resources [1]. At present, the reserves and mining output can no longer meet the growing demand for rare earths [2]. Reutilization of rare earths in waste is an effective way to save rare earth resources [3,4]. Waste fluorescent powder has become a hot spot for rare earth recycling due to its high rare earth content and easy availability [4]. In 2011, the numbers of scrapped fluorescent lamps in the world were about 48 million ton, and the value of rare earth elements contained in them exceeded $1.6 billion [5]. If these rare earth elements can be effectively recovered and reused, not only the mining of primary rare earth ores can be reduced, but also the recycling industry chain of waste rare earth can be established, greatly improving the effective utilization rate of rare earth resources.

Waste phosphors generally contain a mixture of $Y_2O_3$:$Eu^{3+}$ (red phosphors), $CeMgAl_{11}O_{19}$:$Tb^{3+}$ (green phosphors), and $BaMgAl_{10}O_{17}$:$Eu^{2+}$ (blue phosphors) [6]. At present, chemical beneficiation methods such as acid leaching and roasting methods are often used to leach rare earth elements Y and Eu from waste lamp phosphors [7–10] and CRT phosphors [11]. Although these methods can achieve high leaching rates, a longer reaction time and temperature are necessary. In recent years, many researchers have innovatively applied microwave irradiation technology to the hydrometallurgical leaching process, which can improve the leaching rate of useful metals, shorten the leaching reaction times and reduce energy consumption [12,13]. It is usually applied in the extraction process of valuable metals in green, pollution-free, and simplified processes [14,15]. Laubertova [16] et al. indicated that the reaction speed of microwave radiation leaching can be increased by more than two times in the process of recovering zinc and lead from electric arc furnace dusts.

Jenni Lie found that the closed-vessel microwave leaching process proved to be rapid, effective, and efficient for Y and Eu recovery from waste CRT phosphor [17]. However, there are few studies on the leaching of rare earth elements from waste lamp phosphors by microwave radiation. In this paper, the reaction promotion effect caused by microwave radiation was used to achieve the efficient leaching of Y and Eu from waste phosphors.

The mechanism of microwave heating has not been elucidated due to the lack of real-time characterization of microwave irradiation, but the process can be tested by kinetic calculation. Zhang et al. found that microwave heating could significantly enhance the leaching process and improve the leaching rate [18]. Jingpeng Wang et al. found that microwave heating can significantly reduce the activation energy in the reaction process of vanadium leaching from black shale [19]. Yu-kun Huang et al. studied the leaching behavior of mixed rare earth concentrate by microwave heating, and the leaching rate of rare earth elements increased by 16.49% compared with the traditional leaching method [20]. Thiquynhxuan Le found that the leaching kinetics of aluminum and metal impurities from the waste FCC catalyst under microwave radiation followed the core-shrinking model and was controlled by the surface chemical reaction [21]. However, the kinetic analysis of microwave-assisted acid leaching of rare earth elements in waste phosphors was rarely reported. In this paper, microwave heating was used to assist leaching processes, the influence of leaching conditions on the leaching effect of rare earth elements was investigated, and the reaction kinetics of microwave leaching was analyzed.

## 2. Materials and Methods

### 2.1. Waste Phosphors

The raw materials used in this study are the unqualified waste phosphors from Shaanxi fluorescent Material Co., Ltd., China. The chemical composition analysis (Table 1) of the waste phosphors indicates that the content of TREO in the waste phosphors is 49.56%, including 39.80% $Y_2O_3$, 3.14% $Eu_2O_3$, 4.53% $CeO_2$, and 2.16% $Tb_4O_7$. The main non-rare earth component is $Al_2O_3$ with 36.63% content. Its rare earth elements have great recycling value.

**Table 1.** XRF analysis of waste phosphors.

| Component | $Y_2O_3$ | $Al_2O_3$ | BaO | $CeO_2$ | MgO | $Eu_2O_3$ | $Tb_4O_7$ | MnO | $P_2O_5$ |
|---|---|---|---|---|---|---|---|---|---|
| Content (%) | 39.80 | 36.63 | 7.62 | 4.53 | 2.15 | 3.14 | 2.16 | 0.157 | 1.38 |
| Component | SrO | Cl | $Ag_2O$ | $Na_2O$ | $ZrO_2$ | CdO | $SO_3$ | $Fe_2O_3$ | CaO |
| Content (%) | 1.06 | 0.0524 | 0.0491 | 0.0479 | 0.0446 | 0.0291 | 0.0281 | 0.0185 | 0.0858 |

Figure 1 shows the X-ray diffraction pattern of the waste phosphor. It can be seen that the main crystalline phases in the sample are $(Y_{0.95}Eu_{0.05})_2O_3$ (red phosphor, PDF card 25-1011), $(Ce_{0.67}Tb_{0.33})MgAl_{11}O_{19}$ (green phosphor, PDF card 36-0073), and $(Ba_{0.9}Eu_{0.1})Mg_2Al_{16}O_{27}$ (blue phosphor, PDF card 50-0512).

### 2.2. Experimental Methods

A total of 20 g waste fluorescent phosphors was put in the round bottom flask and then mixed with a certain concentration of hydrochloric acid and hydrogen peroxide solution to configurate into a certain concentration of slurry, and a magnetic rotor was put into the round bottom flask. Then the round bottom flask was placed in a microwave reactor. In case of the evaporation of the water in the microwave leaching process, a condenser tube was copped in the top of the round bottom flask. The proper microwave power, reaction time, reaction temperature, and stirring speed were set and then the reaction began. After the reaction was completed, solid-liquid separation was carried out. The content of rare earth ions in the leaching solution was tested by ICP, and the leaching residue was analyzed by X-ray diffraction and X-ray fluorescence. The leaching percentage was calculated using Equation (1).

$$X = (\alpha/\beta) \times 100\% \tag{1}$$

where X is the leaching percentage of Y and Eu, $\alpha$ is the content of Y or Eu in the leach solutions (ICP-OES), and $\beta$ is the contents of Y or Eu in the waste phosphors (XRF).

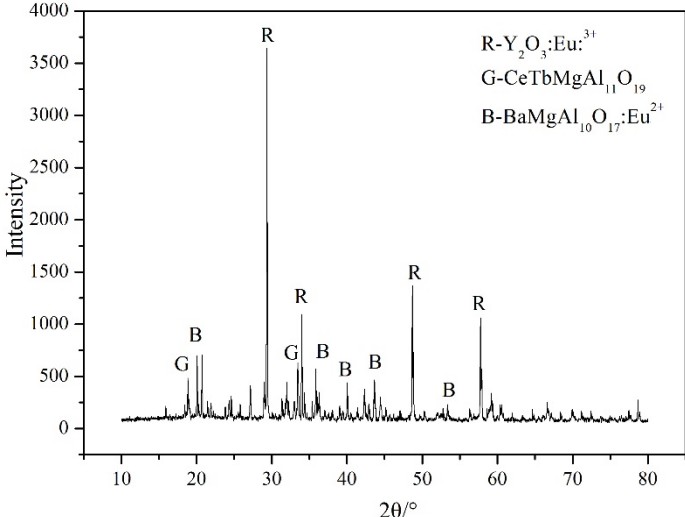

**Figure 1.** XRD analysis of phosphors.

### 2.3. Equipment and Reagents

The microwave leaching reaction was carried out in a microwave reactor (Shanghai Xinyi, MAS-II PLUS, Shanghai, China). The ion content in the leaching solution was determined with an inductively coupled plasma emission spectrometer (ICP-OES, Prodigy7, Mason, OH, USA), the chemical composition of waste phosphors and leaching residues was analyzed with an X-ray fluorescence spectrometer (XRF, AXIOS, Waltham, MA, USA), and the crystalline phase analysis of waste phosphors and leaching residues was analyzed by X-ray diffraction (XRD, Rigaku Dmax-2500, Tokyo, Japan).

## 3. Results

### 3.1. Effect of Microwave Power on the Leaching Rates of Y and Eu

Under the conditions of microwave irradiation time of 120 min, reaction temperature of 50 °C, hydrochloric acid concentration of 4 mol/L, hydrogen peroxide dosage of 0.2 mL/g, and liquid-solid ratio of 10 mL/g, the effects of microwave irradiation power of (400 W, 500 W, 600 W, 700 W, 800 W) on the leaching rates of Y and Eu were investigated and the results are shown in Figure 2.

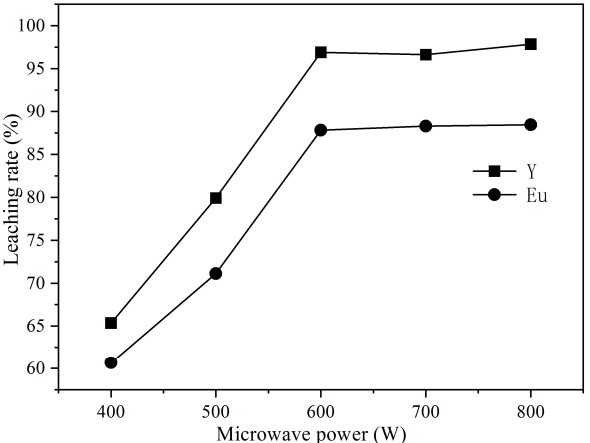

**Figure 2.** Effect of microwave power on the leaching percentage of Y and Eu.

Figure 2 shows that microwave power has a great influence on the leaching rates of Y and Eu. With the increase of microwave power from 400 W to 600 W, the leaching rates of the rare earth elements Y and Eu increase from 65.36% and 60.67% to 96.82% and 87.82%, respectively. This was because with the increase of microwave power, the energy transferred to the leaching system increases, making the temperature of the leaching system rise faster at the same time and strengthening the reaction. When the microwave power was increased to 800 W, the leaching rates of Y and Eu did not increase significantly. Therefore, the optimum microwave power was determined to be 600 W.

### 3.2. The Effect of Reaction Time and Temperature on the Leaching Rates of Y and Eu

The effect of microwave reaction time (20 min, 40 min, 60 min, 80 min, 100 min, and 120 min) and reaction temperature (40 °C, 50 °C, 60 °C, 70 °C, and 80 °C) on the leaching rate of Y and Eu was studied under the following conditions: microwave power of 600 w, microwave reaction time of 60 min, HCl concentration of 4 mol/L, hydrogen peroxide dosage of 0.2 mL/g, and liquid-solid ratio of 7.5:1.

As shown in Figure 3, at the beginning of the reaction, there is enough acid to react with the fluorescent powder, and the reaction rate is faster, so the leaching rate changes significantly. As the reaction time is prolonged, the fluorescent powder consumes a large amount of acid, the acid concentration in the solution decreases, and the easily leached substances have already been leached in the early stage of the reaction, so the leaching rate remains almost unchanged. As the reaction time increased from 20 min to 60 min, the leaching rates of Y and Eu increased from 32.89% and 25.78% to 93.84% and 85.02%, respectively. Further, when increasing the reaction time to 120 min, the leaching rates of Y and Eu did not significantly improve, so the optimal leaching time was selected as 60 min.

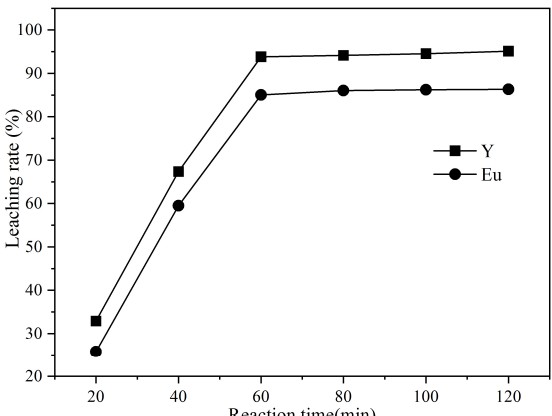

**Figure 3.** Effect of reaction time on the leaching rate of Y and Eu.

According to the experimental results in Figure 4, the increased reaction temperature promotes the decomposition of waste phosphors and the leaching rates of Y and Eu. With the reaction temperature increasing from 40 °C to 60 °C, the leaching rates of Y and Eu increase from 45.21% and 40.78% to 98.84% and 88.72%, respectively, due to the leaching process being an endothermic reaction. At the same time, the increased reaction temperature will speed up the movement of molecules in the leaching solution and accelerate the mass transfer process of the reaction and the dissolution of soluble substances. When the reaction temperature is further increased to 80 °C, the leaching rate of rare earth elements Y and Eu does not increase significantly, and the excessive reaction temperature will accelerate the volatilization of HCl, resulting in environmental pollution. In order to reduce energy consumption and avoid air pollution, 60 °C was selected as the best reaction temperature.

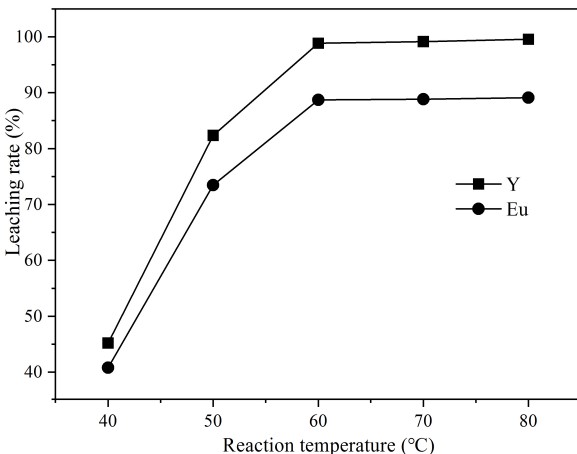

**Figure 4.** Effect of the reaction temperature on the leaching rate of Y and Eu.

*3.3. Effect of the HCl Concentration and $H_2O_2$ Addition on Leaching Rates of Y and Eu*

Under the conditions of microwave power of 600 W, microwave radiation time of 60 min, reaction temperature of 60 °C, and liquid-solid ratio of 7.5 mL/g, the effects of hydrochloric acid concentration (1 mol/L, 2 mol/L, 3 mol/L, 4 mol/L, 5 mol/L, 6 mol/L) and hydrogen peroxide addition amount (0 mL/g, 0.1 mL/g, 0.2 mL/g, 0.3 mL/g, 0.4 mL/g) on the leaching rates of rare earth elements Y and Eu were investigated, and the results are shown in Figures 5 and 6.

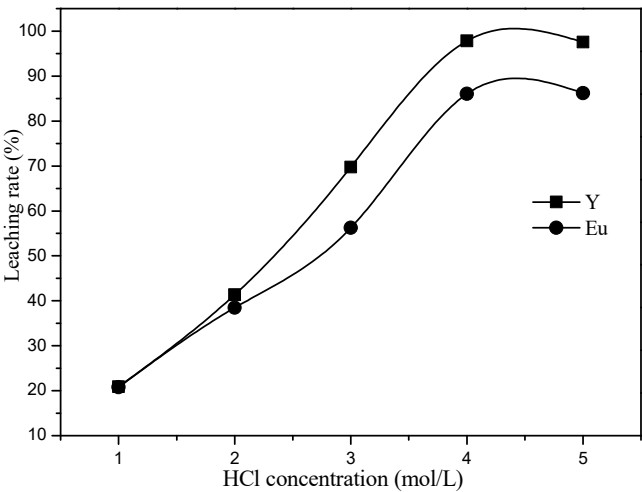

**Figure 5.** Effect of HCl concentration on the leaching percentage of Y and Eu.

As shown in Figure 5, when the concentration of hydrochloric acid increases from 1 mol/L to 4 mol/L, the leaching rates of Y and Eu increase from 20.89% and 20.78% to 97.84% and 86.02%, respectively. The increased hydrochloric acid concentration can improve the leaching rate of rare earth elements. When the concentration of hydrochloric acid is 4 mol/L, the Y and Eu in the red phosphors are almost completely leached. However, due to the stable structure of the blue phosphors, the Eu in the blue phosphors cannot be leached by ordinary acid leaching. When the concentration of hydrochloric acid is increased to 5 mol/L, the leaching rate of Y and Eu does not increase. Therefore, 4 mol/L was considered to be the optimal concentration of hydrochloric acid.

As shown in Figure 6, the addition of hydrogen peroxide significantly improves the leaching rates of Y and Eu, and the leaching rate of Eu is significantly higher than the leaching rate of Y. When the addition of hydrogen peroxide increases from 0 to 0.2 mL/g, the leaching rates of Y and Eu increase from 88.16% and 62.43% to 95.02% and 86.43%, respectively. When the addition of hydrogen peroxide continues to increase, the leaching

rates of Y and Eu did not significantly change. So, the optimal addition of hydrogen peroxide was selected as 0.2 mL/g.

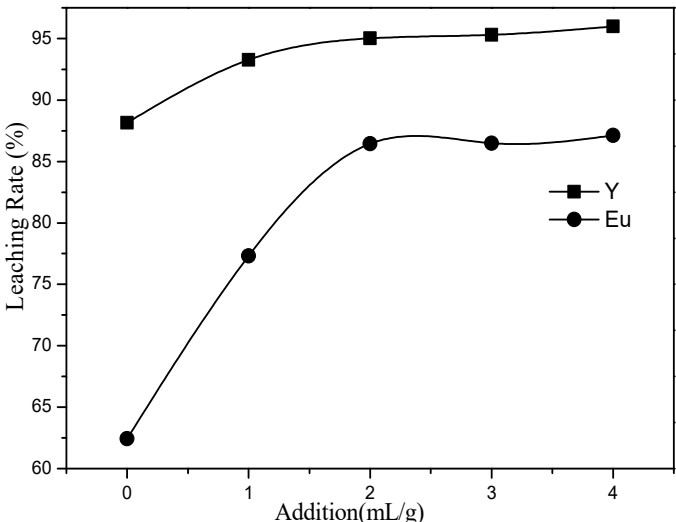

**Figure 6.** Effect of the addition of $H_2O_2$ on the leaching rate of Y and Eu.

*3.4. The Effect of Liquid-Solid Ratio on the Leaching Rate of Y and Eu*

Under the conditions of microwave power of 600 W, microwave radiation time of 120 min, reaction temperature of 60 °C, hydrochloric acid concentration of 4 mol/L, and hydrogen peroxide dosage of 0.2 mL/g, the influence of liquid-solid ratio (2.5 mL/g, 5 mL/g, 7.5 mL/g, 10 mL/g, 12.5 mL/g) on the leaching rates of rare earth elements Y and Eu is shown in Figure 7.

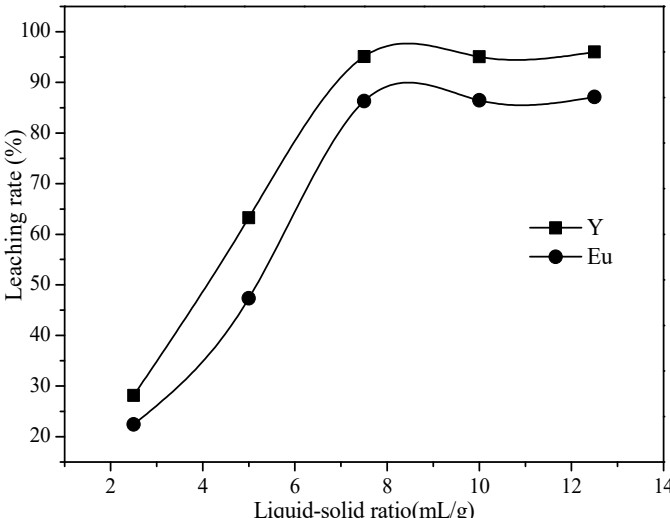

**Figure 7.** Effect of liquid to solid ratio on the leaching rate of Y and Eu.

As shown in Figure 7, as the liquid-solid ratio of the leaching system increased from 2.5:1 to 7.5:1, the leaching rates of Y and Eu increased from 28.16% and 22.43% to 95.11% and 86.32%, respectively. When the liquid-solid ratio was low, there was not enough acid to react with the waste phosphors, resulting in a low leaching rate of rare earth elements. When the liquid-solid ratio of the reaction was high, the acid solution that reacts with the waste phosphors increased, which was conducive to the leaching of rare earth elements. In addition, when the liquid-solid ratio was high, the viscosity of the solution decreased, the mass transfer effect was strengthened, and the dissolution of soluble substances was accelerated, which promoted the reaction. When the liquid-solid ratio was 7.5:1, the

leaching rates of Y and Eu were the highest, and the leaching rates of Y and Eu did not increase significantly when the liquid-solid ratio continued to increase to 12.5:1. Thus, the optimal liquid-solid ratio was selected at 7.5:1.

## 4. The Kinetic Model of the Leaching Reaction

The unreacted shrinking core model can be used in most of the solid-liquid multiple reaction kinetic models for metals. Taking acid leaching of red phosphor as a typical solid-liquid multiple reaction process, it can be determined by the model as follows (Equation (2))

$$A(s) + B(aq) = C(aq) + D(s) \tag{2}$$

According to the shrinking nucleus model, the general steps of the dissolution process were as follows: At first, the reaction leaching agent continuously diffuses to the material surface in the solution, and then the leaching agent diffuses to the material interior after contacting the material surface. Furthermore, the leaching agent reacts with the material surface, and the reaction product diffuses from the reaction boundary to the solid surface. Finally, the product diffuses from the solid surface to the solution. In a micro-perspective, the solid-liquid chemical reaction is generally composed of multiple reaction processes, such as an interface reaction and an internal and external diffusion reaction. Therefore, the leaching rate is generally controlled by the chemical reaction process, the internal diffusion process, and the mixed reaction process of internal diffusion and chemical reaction [22].

Assuming that the phosphor particles have spherical geometry, if the phosphor leaching reaction process is controlled by a chemical reaction, the relationship between the rare earth leaching rate and the leaching time should conform to Equation (3):

$$1 - (1 - x)^{(1/3)} = k_c t \tag{3}$$

where x is the leaching rate of REE in the waste phosphors, and $K_c$ is the chemical reaction rate constant.

If the reaction is controlled by internal diffusion, the relationship between the leaching rate and the leaching time of the rare earth elements should conform to Equation (4):

$$1 - (2/3)x - (1 - x)^{(2/3)} = k_d t \tag{4}$$

where $K_d$ is the rate constant of the internal diffusion process.

If the reaction is controlled by the mixed reaction process of internal diffusion and chemical reaction, the relationship between the leaching rate and the leaching time of the rare earth elements should conform to Equation (5):

$$1/3\ln(1 - x) + (1 - x)^{-(1/3)} = k_e t \tag{5}$$

where $K_e$ is the mixed controlled reaction rate constant.

### 4.1. The Leaching Reaction Apparent Activation Energy of Y and Eu

It is assumed that the waste phosphor particles are approximately spherical, and the leaching agents in the solution are hydrochloric acid and hydrogen peroxide. The experimental data were evaluated through the shrinkage core model of internal diffusion control and chemical reaction control and mixed control used during the leaching process of the waste phosphors. Under different reaction temperatures, the $1 - (1 - x)^{(1/3)}$, $1 - (2/3)x - (1 - x)^{(2/3)}$, $1/3 \ln(1 - x) + (1 - x)^{-(1/3)}$ and time will be distributed in a straight line, where the straight line slope is the apparent reaction rate constant. According to the coefficient of determination ($R^2$) between the experimental data and the dynamics model, the possibility of the dynamics model can be judged as shown in Table 2.

According to the data in Table 2, the model of chemical reaction control $1 - (1 - x)^{(1/3)}$ has the highest fitting degree. Therefore, the leaching process of Y and Eu conforms to the model of chemical reaction control.

**Table 2.** Model Fitting of Rare Earth Element Y and Eu.

| T (K) | | Correlation Value ($R^2$) | | | | |
|---|---|---|---|---|---|---|
| | | Y | | | Eu | |
| | $1 - (1 - x)^{(1/3)}$ | $1 - (2/3)x - (1 - x)^{(2/3)}$ | $1/3\ln(1 - x) + (1 - x)^{-(1/3)}$ | $1 - (1 - x)^{(1/3)}$ | $1 - (2/3)x - (1 - x)^{(2/3)}$ | $1/3\ln(1 - x) + (1 - x)^{-1/3}$ |
| 323 | 0.9922 | 0.9401 | 0.8773 | 0.9969 | 0.8744 | 0.7023 |
| 333 | 0.9980 | 0.9575 | 0.8656 | 0.9971 | 0.8827 | 0.6769 |
| 343 | 0.9917 | 0.9470 | 0.6681 | 0.9948 | 0.8869 | 0.6753 |
| 353 | 0.9921 | 0.7851 | 0.6938 | 0.9963 | 0.8124 | 0.6597 |
| 363 | 0.9958 | 0.9825 | 0.7362 | 0.9952 | 0.7469 | 0.6232 |

In order to determine the influence of hydrochloric acid concentration, temperature, and other variable factors on hydrochloric acid leaching kinetics of waste phosphors, the following semi-empirical model was established:

$$1 - (1 - x)^{(1/3)} = k_0 \times C_{(HCl)}^a \times e^{-Ea/(RT)} \times t \tag{6}$$

$$k_c = k_0 C_{(HCl)}^a \times e^{-Ea/(RT)} \tag{7}$$

where t is the temperature, $k_0$ is the constant of the apparent rate of the reaction, $C_{(HCL)}$ is the concentration of the HCl, and Ea is the apparent activation energy of the reaction.

When only the temperature is changed, the equation can be written as follows:

$$1 - (1 - x)^{(1/3)} = k_0 e^{-Ea/(RT)} \times t \tag{8}$$

$$k_c = k_0 e^{-Ea/(RT)} \tag{9}$$

Take the logarithm of both sides of the above equation:

$$\ln[k_c] = \ln k_0 - Ea/(RT) \tag{10}$$

According to the above formula, the apparent activation energy Ea of Y and Eu in the leaching process can be calculated by making the Arrhenins diagram.

Figures 8 and 9 show the relationship between the Y and Eu leaching rate and leaching time at different leaching temperatures under microwave irradiation. It can be seen that with the increase in leaching temperature, the leaching rate of Y and Eu increases continuously. According to the data shown in the figure, Equation (7) is used to fit the data, and the fitting results are shown in Figures 10 and 11.

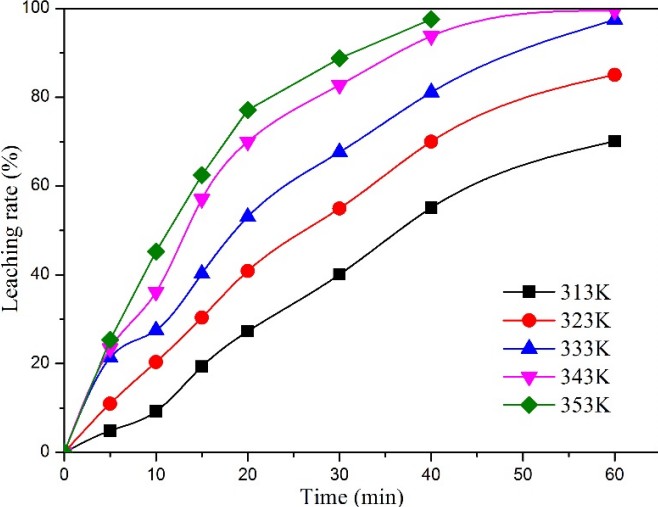

**Figure 8.** Effect of reaction time on the leaching rate of Y at different reaction temperature.

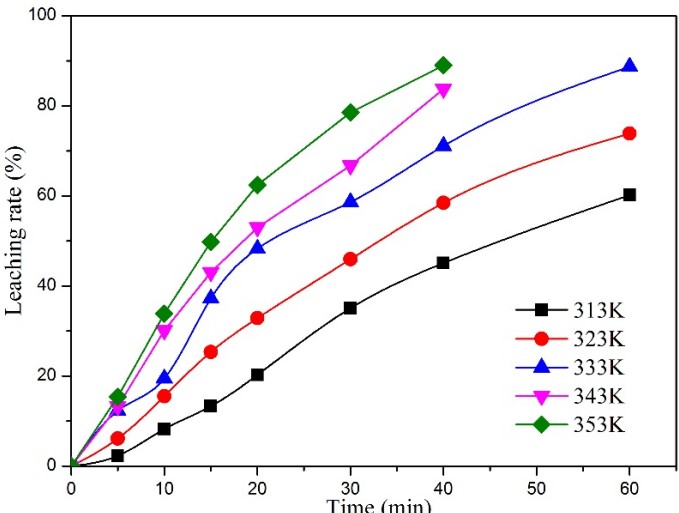

**Figure 9.** Effect of reaction time on the leaching rate of Eu at different reaction temperature.

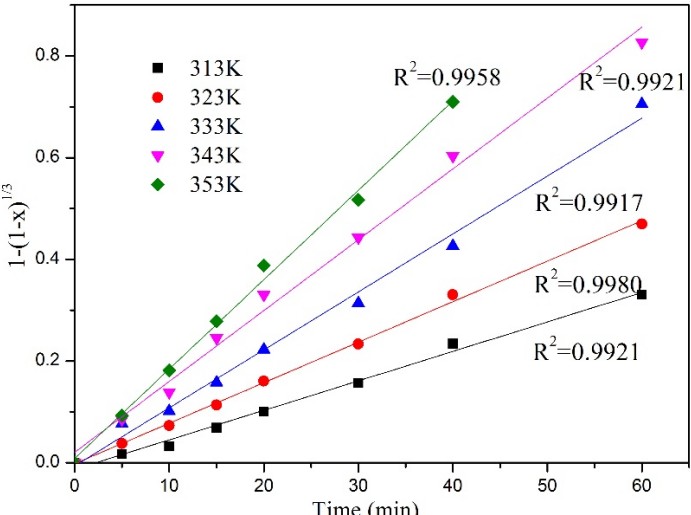

**Figure 10.** Chemical controlled kinetic model of Y leaching rate at different temperatures.

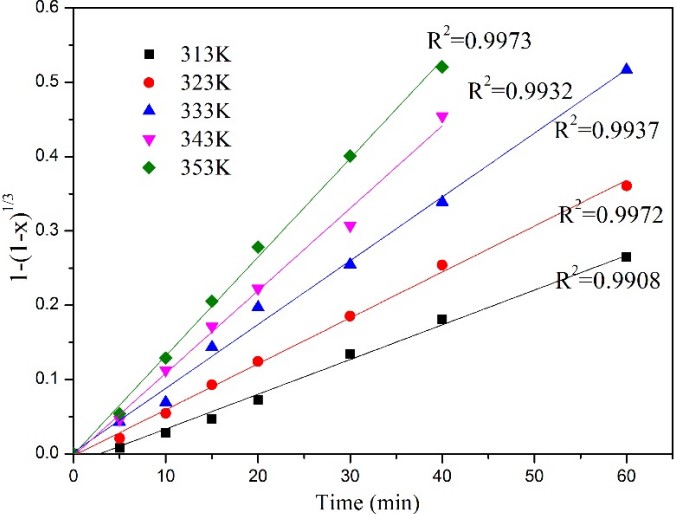

**Figure 11.** Chemical controlled kinetic model of Eu leaching rate at different temperatures.

The values of the specific rate constants obtained from the kinetic plots were used to construct the Arrhenius plot (Figure 12). The activation energy (Ea) for the leaching of Y and Eu has been calculated to be 25.304 and 24.78 kJ/mol kJ/mol, respectively. Although the apparent activation energy was less than 40 kJ/mol, the high fitting degree indicated that the leaching process corresponded to the chemical reaction control model.

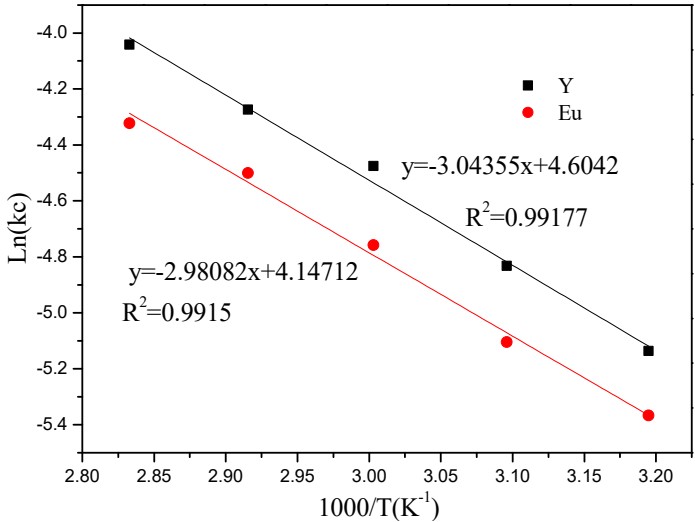

**Figure 12.** Arrhenius plot for leaching of Y and Eu.

### 4.2. Comparison of Microwave Leaching and Conventional Leaching

Conventional leaching was carried out in a thermostatic oil bath with magnetic stirring and its conditions were as follows: hydrochloric acid concentration of 4 mol/L, hydrogen peroxide dosage of 0.2 mL/g, liquid-solid ratio of 7.5:1, reaction temperature of 60 °C, and liquid-solid ratio of 7.5:1 [7]. Microwave leaching was the optimal experimental condition above, and the experimental comparison results were shown in Figure 10.

It can be seen from Figure 13 that the leaching rates of the rare earth elements Y and Eu reach the maximum when the reaction time is 180 min under the traditional heating method, and the leaching rates are 98.36% and 88.32%, respectively. Compared with the traditional heating method, microwave leaching can greatly reduce the reaction time; 60 min of reaction time can achieve the maximum leaching rate, and the leaching rate of Y and Eu were 98.84% and 88.72%, respectively. The results indicate that microwave leaching can greatly reduce the reaction time without reducing the leaching rate.

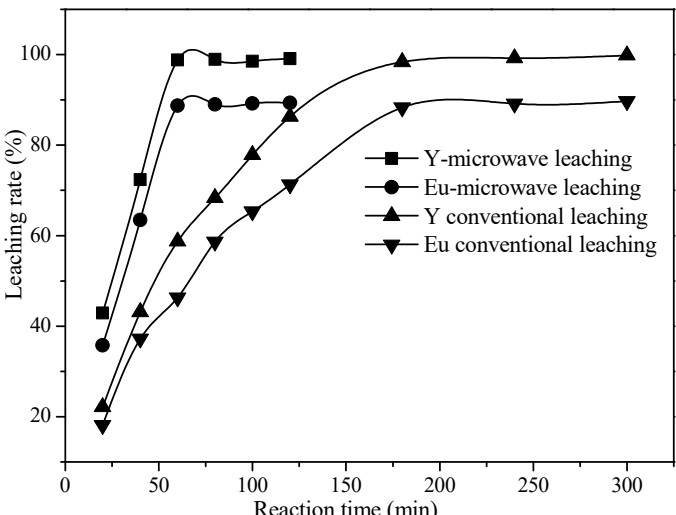

**Figure 13.** Comparison of leaching rates between microwave heating and traditional heating.

By comparing the leaching effect and apparent activation energy under different leaching methods [7], it was found that microwave heating can reduce the activation energy of the leaching reaction in the leaching process and enhance the leaching of the rare earth elements Y and Eu in the waste phosphors, resulting in a shorter leaching time.

### 4.3. Phase Changes of Waste Phosphors before and after Leaching

Phase analysis of waste phosphors and leaching residue before and after microwave leaching is shown in Figure 14.

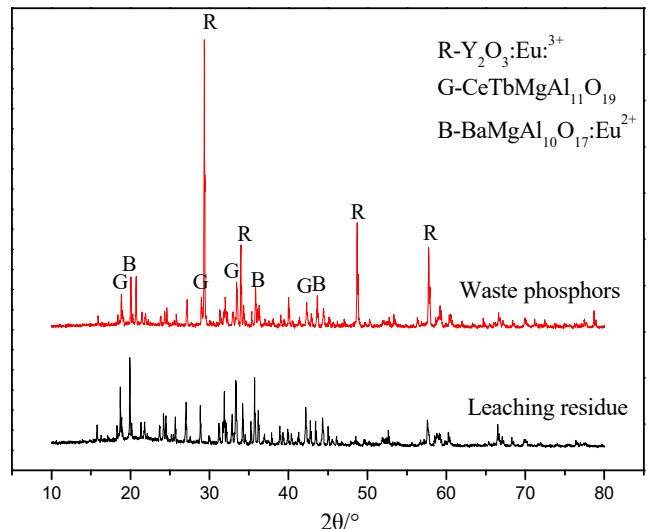

**Figure 14.** XRD analysis of waste phosphors and leaching residue.

According to Figure 14, compared with the waste phosphors before leaching, the peak of $(Y_{0.95}Eu_{0.05})_2O_3$ (red phosphor) in the leaching residue almost disappeared, indicating that the red phosphor was almost completely leached during the leaching process. The main phase of leach residue is $(Ce_{0.67}Tb_{0.33})MgAl_{11}O_{19}$ (green phosphor) and $(Ba_{0.9}Eu_{0.1})Mg_2Al_{16}O_{27}$ (blue phosphor), and the peak increases correspondingly, indicating that the leach residue is mainly green and blue phosphors. The result of XRD shows that the microwave leaching using hydrochloric acid can preferentially leach red phosphor from the waste phosphors and achieve the selective leaching of Y and Eu.

### 5. Conclusions

The maximum leaching rate of Y and Eu (99.84% and 88.02%) can be obtained at a microwave power of 600 W, microwave radiation time of 60 min, and reaction temperature of 60 °C, which achieved the preferential leaching of rare earth elements Y and Eu in the waste phosphors. Microwave radiation-assisted leaching can obviously shorten the reaction time without reducing the leaching rate.

The leaching process of rare earth elements Y and Eu conforms to the chemical reaction control model, and their apparent activation energies are 25.304 kJ/mol and 24.78 kJ/mol, respectively. The microwave leaching process can obviously reduce the reaction activation energy of the leaching process.

**Author Contributions:** D.Y. performed experiments, modeling, and simulation and wrote this paper. M.C. and Y.Z. assisted with laboratory and piloting equipment as well as with expert knowledge regarding operations and objectives. M.Y. and G.M. was responsible for conception and supervision. All authors have read and agreed to the published version of the manuscript.

**Funding:** This work was financially supported by the "National Natural Science Foundation of China" (No: 51904121), the Natural Science Foundation of the Jiangxi Province (No. 20224BAB204038, 20202BAB214014), the Young Elite Scientists Sponsorship Program by CAST (2022QNRC001), the opening project of Jiangxi University of Science and Technology on the development and utilization of rare earth resources jointly established by the province and the ministry (JXUST-XTCX-2022-05), and the Open Project of Guangxi Key Laboratory of Nonferrous Metals and Characteristic Materials Processing (2022GXYSOF11).

**Data Availability Statement:** Data cannot be made publicly available.

**Conflicts of Interest:** The authors declare no conflict of interest.

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
