# Peer review of "Leaching Kinetics of Y and Eu from Waste Phosphors under Microwave Irradiation"

_processes, doi:10.3390/pr11071939_

Round 1
Reviewer 1 Report
The authors have presented on the kinetics of leaching Y and Eu from waste phosphors via microwave irradiation. The manuscript has been well written but has some few technical issues which need to be addressed to improve the quality of the manuscript. The following are some few issues which need to be addressed.
1. The authors need to define acronyms before they are used in the manuscript (e.g CRT, FCC)
2. A table to summarise the chemical composition of the waste phosphors should be provided.
3. Why did the authors choose to use XRF to determine leaching rate of rare earth elements in solids/residues?
4. The authors are encouraged to use straight lines instead of curved/smooth lines) in the graphs (Figures 2 and 4)
5. Line 138: Check the leaching rate values, they do not tally with those in Figure 3.
6. Line 193: there is no Table 2 in the manuscript, please check/correct this
7. Line 250: Are the authors referring to Figure 11, as the manuscript does not show Figure 12?
The paper needs further proof reading to correct some few grammatical errors
Author Response
Dear Review,
Thank you very much for your reply and help. We provide this cover letter to explain, point by point, the details of our revisions in the manuscript and our responses to the reviewer.
1.The authors need to define acronyms before they are used in the manuscript (e.g CRT, FCC)
The acronyms have been defined.
2.A table to summarise the chemical composition of the waste phosphors should be provided.
Table1 XRF analysis of waste phosphors has been provided.
3. Why did the authors choose to use XRF to determine leaching rate of rare earth elements in solids/residues?
The leaching percentage was calculated using Eq (1).
X=(α/β)×100% (1)
Where X is the leaching percentage of Y and Eu, α is the content of Y or Eu in the leach solutions (ICP-OES) and β is the contents of Y and Eu in the waste phosphors (XRF).
4. The authors are encouraged to use straight lines instead of curved/smooth lines) in the graphs (Figures 2 and 4)
The Figures 2, 3and 4 has been changed.
5.Line 138: Check the leaching rate values, they do not tally with those in Figure 3.
The data has been confirmed and modified.
6. Line 193: there is no Table 2 in the manuscript, please check/correct this.
The table order has been confirmed and modified.
7.Line 250: Are the authors referring to Figure 11, as the manuscript does not show Figure 12?
The Figure 12 should be Figure 11.
Reviewer 2 Report
The authors have studied the leaching kinetics of two rare earth elements from a secondary source. Some ambiguity and concerns are listed below:
1-Considering the high corrosive nature of the applied reagents, is there any hope for industrial application of f hydrochloric acid and hydrogen peroxide as the main lixiviants in future? Is there any commercial system right now working with such reagents and such concentration?
2-How is it possible to keep a certain temperature while microwave power is different? What does microwave do without heating? It seems that you have reported the surface temperature while the microwave changes the internal temperature.
3-Have you measured the evaporation of reagents in different temperature?
4-Parameter R2 is not a correlation factor. It is called coefficient of determination and is the proportion of the variation in the dependent variable that is predictable from the independent variable(s).
5-R2 can not be compared between several models and determine which model is "phenomenologically" best suited. You can compare the diffusion model's R2 with a worst R2 model and claim that diffusion is the model!! Selection of a model should be based on real physics and chemistry of the system, and not on a statistical value. A shrinking core leave ashes of unreacted material, while a shrinking particle does not leave such residue. Why you have not chosen a shrinking particle method?
6-It is stated that "it was found that microwave heating can reduce the activation energy of leaching reaction in the leaching process". You have not reported the activation energy of the leaching process without microwave assist. To what basis are comparing your results?
The quality of English language is good.
Author Response
Dear Reviewer,
Thank you very much for your reply and help. We provide this cover letter to explain, point by point, the details of our revisions in the manuscript and our responses to the reviewer.
1-Considering the high corrosive nature of the applied reagents, is there any hope for industrial application of hydrochloric acid and hydrogen peroxide as the main lixiviants in future? Is there any commercial system right now working with such reagents and such concentration?
At present, the most commonly used acid for acid leaching in industry is hydrochloric acid, and hydrochloric acid has been widely used in the recycling of waste batteries and waste permanent magnets.
2-How is it possible to keep a certain temperature while microwave power is different? What does microwave do without heating? It seems that you have reported the surface temperature while the microwave changes the internal temperature.
The microwave oven used in this experiment can set the reaction temperature and operate under constant temperature conditions.
This microwave oven can use a temperature sensing probe or infrared to measure the temperature inside the solution.
3-Have you measured the evaporation of reagents in different temperature?
Evaporation has not been tested. However, the microwave oven can use a condensation reflux device, so evaporation has little impact on the test.
4-Parameter R2 is not a correlation factor. It is called coefficient of determination and is the proportion of the variation in the dependent variable that is predictable from the independent variable(s).
The R2 has been modified to coefficient of determination.
5-R2 can not be compared between several models and determine which model is "phenomenologically" best suited. You can compare the diffusion model's R2 with a worst R2 model and claim that diffusion is the model!! Selection of a model should be based on real physics and chemistry of the system, and not on a statistical value. A shrinking core leave ashes of unreacted material, while a shrinking particle does not leave such residue. Why you have not chosen a shrinking particle method?
The experimental data were evaluated by the shrinkage core model of internal diffusion control and chemical reaction control and mixed control used during the leaching process of waste phosphors.
6-It is stated that "it was found that microwave heating can reduce the activation energy of leaching reaction in the leaching process". You have not reported the activation energy of the leaching process without microwave assist. To what basis are comparing your results?
References have been added here. The reference literature studied the leaching kinetics under conventional heating conditions.
Reviewer 3 Report
the authors made an attempt on Leaching kinetics of Y and Eu from waste phosphors under mi- 2 crowave irradiation. However A major revision ir required to enhance the quality of manuscript
Abstract needs to be quantified with scientific values
Literatures related to previous findings needs to improved from that the research gap needs to be identified
the novelty pf present investigation needs to be improved
Experimentation is not sufficient. Need to discuss more with levant findings along with experimentation photography
Results and discussion needs to improved with similar findings
the authors made an attempt on Leaching kinetics of Y and Eu from waste phosphors under mi- 2 crowave irradiation. However A major revision ir required to enhance the quality of manuscript
Abstract needs to be quantified with scientific values
Literatures related to previous findings needs to improved from that the research gap needs to be identified
the novelty pf present investigation needs to be improved
Experimentation is not sufficient. Need to discuss more with levant findings along with experimentation photography
Results and discussion needs to improved with similar findings
Author Response
Dear Reviewer,
Thank you very much for your reply and help. We provide this cover letter to explain, point by point, the details of our revisions in the manuscript and our responses to the reviewer.
(1) Abstract needs to be quantified with scientific values.
Abstract has been changed.
(2) Literatures related to previous findings needs to improved from that the research gap needs to be identified.
The introduction has been revised.
(3) the novelty of present investigation needs to be improved.
The introduction has been revised to improve the novelty of this paper.
(4) Experimentation is not sufficient. Need to discuss more with levant findings along with experimentation photography.
In addition to the microwave power, microwave time, and microwave temperature mentioned in the paper, experiments were also conducted on hydrochloric acid concentration, liquid-solid ratio, and H2O2 addition amount. Because the trend of changes in these experiments is not significant compared to conventional leaching, so the references are provided in the section of 2.2. Experimental methods, and it is not explained in detail in the results and discussion section.
The discussion on figure has also been appropriately added
(5) Results and discussion needs to improved with similar findings.
The results and discussion have been changed.
Round 2
Reviewer 2 Report
The role of microwave (heating or crack generation), the role of internal temperature and how it was maintained constant, the choose of model based on experimental results and other questions asked in first round of review are not addressed completely in the responses of authors.
Author Response
The role of microwave:
In this experiment, microwave was used as a heating method. The occurrence of cracks is not currently studied in this article
The role of internal temperature and how it was maintained constant:
This experiment measures the internal temperature of the slurry during the reaction process using an infrared temperature measuring device built-in in the reaction equipment. The experimental equipment used in this experiment can control the reaction temperature. If the temperature exceeds the set temperature, microwave heating can be stopped. If the temperature is lower than the set temperature, microwave radiation begins, similar to a constant temperature water bath.
The choose of model based on experimental results;
The most commonly used reaction model for solid-liquid reaction processes is the unreacted nuclear contraction model. As for the shrinkage particle model, there has been less attention previously, and we will carefully compare the advantages and disadvantages of these two models in the future.
Other questions asked in first round:
(1)Considering the high corrosive nature of the applied reagents, is there any hope for industrial application of hydrochloric acid and hydrogen peroxide as the main lixiviants in future? Is there any commercial system right now working with such reagents and such concentration?
At present, the most commonly used acid for acid leaching in industry is hydrochloric acid, and hydrochloric acid has been widely used in the recycling of waste batteries and waste permanent magnets.
2-How is it possible to keep a certain temperature while microwave power is different? What does microwave do without heating? It seems that you have reported the surface temperature while the microwave changes the internal temperature.
The experimental equipment used in this experiment can control the reaction temperature. If the temperature exceeds the set temperature, microwave heating can be stopped. If the temperature is lower than the set temperature, microwave radiation begins, similar to a constant temperature water bath.
In this experiment, microwave was used as a heating method. The occurrence of cracks is not currently studied in this article.
This experiment measures the internal temperature of the slurry during the reaction process using an infrared temperature measuring device built-in in the reaction equipment.
3-Have you measured the evaporation of reagents in different temperature?
Evaporation has not been tested. However, the microwave oven can use a condensation reflux device, so evaporation has little impact on the test.
4-Parameter R2 is not a correlation factor. It is called coefficient of determination and is the proportion of the variation in the dependent variable that is predictable from the independent variable(s).
The R2 has been modified to coefficient of determination.
5-R2 can not be compared between several models and determine which model is "phenomenologically" best suited. You can compare the diffusion model's R2 with a worst R2 model and claim that diffusion is the model!! Selection of a model should be based on real physics and chemistry of the system, and not on a statistical value. A shrinking core leave ashes of unreacted material, while a shrinking particle does not leave such residue. Why you have not chosen a shrinking particle method?
The experimental data were evaluated by the shrinkage core model of internal diffusion control and chemical reaction control and mixed control used during the leaching process of waste phosphors.
The most commonly used reaction model for solid-liquid reaction processes is the unreacted nuclear contraction model. As for the shrinkage particle model, there has been less attention previously, and we will carefully compare the advantages and disadvantages of these two models in the future.
6-It is stated that "it was found that microwave heating can reduce the activation energy of leaching reaction in the leaching process". You have not reported the activation energy of the leaching process without microwave assist. To what basis are comparing your results?
References have been added here. The reference literature studied the leaching kinetics under conventional heating conditions.
Reviewer 3 Report
The research carried out research on Leaching kinetics of Y and Eu from waste phosphors under mi- 2 crowave irradiation. however a major correction was given to improve the quality of manuscript
upon viewing the researches was carried out all correction and it was satisfied. So that it may accept in the present form and may be forwarded to further process.
Author Response
Thank you for your support and wish you success in your work.